# Pipeline Elbow Corrosion Simulation for Strain Monitoring with Fiber Bragg Gratings

**DOI:** 10.3390/mi15091098

**Published:** 2024-08-29

**Authors:** Kaimin Yu, Zixuan Peng, Yuanfang Zhang, Peibin Zhu, Wen Chen, Jianzhong Hao

**Affiliations:** 1School of Marine Equipment and Mechanical Engineering, Jimei University, Xiamen 361021, China; 202212855006@jmu.edu.cn; 2School of Ocean Information Engineering, Jimei University, Xiamen 361021, China; 202121301036@jmu.edu.cn (Z.P.); 202311810006@jmu.edu.cn (Y.Z.); peibin.zhu@jmu.edu.cn (P.Z.); 3Institute for Infocomm Research (I^2^R), Agency for Science, Technology and Research (A⋆STAR), Singapore 138632, Singapore

**Keywords:** optical fiber sensor, corrosion, non-destructive testing, strain, pipe elbow

## Abstract

This study addresses the limitation of traditional non-destructive testing methods in real-time corrosion monitoring of pipe elbows by proposing the utilization of fiber Bragg grating (FBG) strain sensors, renowned for their resilience in harsh environments. However, the current mathematical relationship model for strain representation of elbow corrosion is still lacking. This paper develops a finite element model to scrutinize the strain changes in the elbow due to corrosion under hydrostatic pressure and bending loads. To mitigate temperature loading effects, the corrosion degree is evaluated through the disparity between hoop and axial strains. Simulation outcomes reveal that, under hydrostatic pressure, the strain difference exhibits minimal changes with the increase in corrosion degree, while under bending moment loading, the strain difference escalates proportionally with corrosion progression. Consequently, strain induced by bending moment loading solely characterizes the corrosion degree. Moreover, the optimal placement for FBG sensors is identified at the extrados of the pipe elbow, where strain is most prominent. These insights enhance comprehension of strain–corrosion dynamics in pipe elbows, offering valuable guidance for developing an FBG-based monitoring system for real-time corrosion tracking and predictive maintenance of pipeline infrastructures.

## 1. Introduction

Metal piping is widely used as critical infrastructure in areas such as the aviation, transportation, petrochemical, food and pharmaceutical, automotive, and nuclear industries [1,2,3]. Pipe elbows are an important part of the piping system and are used to change the direction of water flow. However, factors such as corrosive substances in the fluid, solid particles, high flow rates, elevated temperatures and pressures, and inappropriate material selection often lead to continuous corrosion and thinning of pipe elbows [4,5,6,7,8,9,10].

The wall thickness of pipe elbows gradually thins during continuous operation, which may eventually lead to their bursting or to the leakage of hazardous substances, resulting in economic losses, environmental pollution, and personal safety problems [11]. Despite significant advancements in materials science and chemical technologies—such as deoxygenation, corrosion potential control, and pH regulation—and the development of corrosion-resistant materials like SS310, Ti-23Nb alloy, and Alloy 740, completely preventing corrosion remains unattainable [12,13,14,15,16]. Therefore, it is necessary to monitor the elbow arch back thickness in real time to forecast and warn of accidents in advance. Conventional pipeline corrosion level monitoring techniques include non-destructive testing (NDT) techniques such as microwave testing (MT), infrared thermography (IRT), ultrasonic testing (UT), digital radiography (DR), pulsed eddy current testing (PECT), and vibration-based structural damage identification (VBSDI) [17,18,19,20,21,22,23,24]. However, MT is strongly influenced by electromagnetic parameters and material geometry, limiting its effectiveness in detecting small thicknesses or metal cladding, and its signal is susceptible to interference. IRT, as an indirect detection method, focuses on cladding defects and moisture but struggles to directly reflect the health of the pipe. When dealing with complex structures such as cladding and elbows, UTs can encounter serious problems with sound scattering and energy absorption. The location of defects can also be greatly affected by the angle of incidence. DR provides a visual image but has a limited detection range and requires stringent radiation safety controls. PECT is currently only effective for small-diameter pipes due to the concentration of energy in the non-vortex-induced DC component and the low-frequency harmonics of the weak vortex field, resulting in weak signals, low signal-to-noise ratios, and a limited depth of detection. VBSDI utilizes frequency variations to monitor pipeline conditions, but accurate frequency extraction is susceptible to interference from complex environments and structures [25,26]. Recent research introduced a VBSDI-based approach to assess the thinning state of pipe elbows with internal fluids by extracting modal frequencies [27]. However, this method does not pinpoint the exact location of thinning or quantitatively measure the elbow wall’s thickness change, and its limited sensitivity hampers the detection of minor wall thinning [28]. In short, the existing elbow thickness monitoring methods are either technically complex, require expensive equipment, or are unable to quantitatively characterize the thickness of the pipe wall. More importantly, they are unable to perform real-time monitoring under high temperature and pressure conditions. Therefore, there is an urgent need for cost-effective, real-time, quantitative monitoring of pipe wall thickness under high temperature and pressure conditions.

Fiber optic sensing technology has the advantages of simple operation, high sensitivity and accuracy, light weight, strong distributed measurement ability, anti-electromagnetic interference, high temperature and high pressure, acid and alkali environments, etc. [29,30,31,32], so that it can be used to capture subtle strain changes in real time and diagnose the degree of slight thinning of the pipeline at the beginning of the harsh environment. It is divided into three main categories [29,32,33,34,35,36,37,38,39,40,41,42,43]: point, quasi-distributed, and distributed fiber optic sensors. Point-type fiber optic sensors, such as enamel cavity fiber optic sensors or fiber grating sensors, commonly use the change of light signals reflected from the end face of the optical fiber to reflect back the change in environmental parameters. It has the advantages of high sensitivity, strong anti-interference ability, and easy remote monitoring but can only measure changes in environmental parameters, cannot obtain absolute parameter values, and cannot be multiplexed, limiting it to monitoring a single point. Distributed fiber optic sensors, such as the Brillouin optical time domain reflectometer (BOTDR), the Brillouin optical time domain analyzer (BOTDA), and optical frequency domain reflectometry (OFDR), can continuously monitor an entire pipeline up to hundreds of kilometers long, but its measurement accuracy is relatively low and limited by the spatial resolution and measurement distance. Quasi-distributed fiber optic sensors, such as fiber Bragg grating (FBG) sensors, are widely used in corrosion detection along straight pipe sections due to their high accuracy, sensitivity, spatial resolution, and multipoint monitoring abilities [44,45,46].

However, regardless of which fiber optic sensor is used, it is necessary to first determine the relationship between the optical sensing signal and the degree of corrosion. Three types of methods are commonly used, including analytical solution methods, experimental calibration methods, and numerical methods. Analytical methods, while direct and straightforward, can face challenges in deriving formulas for representing the corrosion degree in complex scenarios such as unevenly corroded bends [47,48]. Experimental calibration involves meticulous control of loading conditions in the lab, tracking the Bragg wavelength drift as corrosion progresses. However, this method may not fully capture the diverse bends of different materials and structures [49]. On the other hand, finite element analysis (FEA), a numerical approach, proves valuable in simulating physical field distributions in corroded bends of varying materials and complex structures under different loading conditions. FEA enables the exploration of the correlation between the corrosion level and the Bragg wavelength with a higher degree of flexibility and accuracy [50]. Recently, an FEA reported the variation of strain monitored by FBG sensors with the bending of thin plates of different thickness [51], which proved that strain can be used to characterize the thickness variation of thin plates. Thus, quasi-distributed FBG sensors are the most promising application for corrosion monitoring of pipe elbows. However, the thin plate structure is significantly different from the pipe bending structure, especially under complex loading conditions, such as internal fluid pressure fluctuations, drastic temperature changes, and bending moment loads, etc., and the strain distribution of metal pipe elbows with complex structural shapes is complicated, especially in terms of different degrees of non-uniform corrosion of the pipe wall. Therefore, in order to be able to quantitatively measure the degree of corrosion of pipe elbows using grating-monitored elbow strains, it is necessary to study the strain changes of pipe elbows with different degrees of corrosion under complex loads.

This work builds a finite element model (FEM) of a metal pipe elbow with the aim of investigating its strain characteristics under complex loading conditions during corrosion. First, the principle of FBG sensors to monitor the corrosion of metal bends is described. Secondly, in order to understand the corrosion process of metal pipe elbows, the strain distribution of the arch back of metal pipe elbows with different thicknesses under hydrostatic internal pressure and bending moment loading, as well as the strain changes caused by the collapse of the pipe elbow structure, are respectively simulated. Finally, the principle of mechanical superposition is used to obtain the relationship between the strain change of the arch back of the elbow and its wall thickness under the composite load. The areas of greatest strain change due to increased corrosion are discussed, and the optimum location for installing the FBG sensors is determined accordingly. This study reveals the multidimensional factors affecting the strain change of the pipe elbow, establishes the relationship between the strain of the elbow and its corrosion degree, and lays the theoretical foundation for future experiments.

## 2. Principle of Measuring the Wall Thickness by FBG Corrosion Sensor

The principle of monitoring the wall thickness of a metal pipe elbow with an FBG sensor is shown in Figure 1.

Considering that the bent pipe is subjected to bending moment load, its radius of curvature will change, as shown in Figure 1a. The inside of the pipe axis is compressed, while the outside of the axis is stretched. Since the arch back of the bend is prone to corrosion thinning, we focus on the strain changes in this region, as shown in Figure 1b. The outer side of the bend can be considered as a narrow fiber beam parallel to its axis. At any given cross-section, the region of fibers near the convex side will be stretched, while the concave side will be compressed. Since the stresses in the material must be continuous in any cross-section, there must be a boundary, namely a neutral surface, between the stress-free fiber compression and tension regions. When a beam of thickness 2h undergoes elastic deformation with a radius of curvature of ρ+h, as shown in Figure 1b, the tensile strain in the convex portion is (ρ+h)dθ. If the inner wall is thinned by corrosion (blue line moving towards red line), the tensile strain is (ρ+h−Δh)dθ. The axial strain on the elongated surface can be expressed as:(1)$$\varepsilon_{a}=\frac{(\rho +h-\Delta h)d\theta -(\rho +h)d\theta }{(\rho +h)d\theta }=-\frac{\Delta h}{\rho +h},$$
where the negative sign indicates the compressive strain. The hoop strain at the same position can be obtained according to Hooke’s law:(2)$$\varepsilon_{h}=-\nu \varepsilon_{a}=\frac{\nu \Delta h}{\rho +h},$$
where ν is the Poisson ratio. Combining Equations (Equation 1) and (Equation 2), the thickness change of the pipe elbow wall can be expressed as:(3)$$\Delta h=\left(\rho +h\right)\left(\frac{1}{\nu }+1\right)\left(\varepsilon_{h}-\varepsilon_{a}\right).$$

Therefore, it is possible to estimate the degree of uniform corrosion by measuring the strain on the convex surface, which can be realized by means of an FBG sensor. The strain and temperature change on the concave surface of a pipe elbow can be obtained with a Bragg wavelength shift Δλ:(4)$$\frac{\Delta \lambda }{\lambda }=k\varepsilon +\left(\alpha_{\Lambda }+\alpha_{n}\right)\Delta T,$$
where λ is the initial Bragg wavelength, *k* is the strain coefficient at the Bragg wavelength of the fiber, $\alpha_{\Lambda }$ is the coefficient of thermal expansion, $\alpha_{n}$ is the coefficient of thermo-optics, and ΔT is the temperature change. The first term represents the strain-induced shift in the Bragg wavelength, and the second term represents the temperature-induced shift. Therefore, the FBG sensor can directly measure strain and temperature. When calculating the degree of corrosion using Δλ, substitute Equation (Equation 4) into Equation (Equation 3). Since the pipe elbow is isotropic, its $\alpha_{\Lambda }$ and $\alpha_{n}$ are uniform in all directions. Therefore, by correcting for temperature-induced wavelength shift using $\left(\varepsilon_{h}-\varepsilon_{a}\right)$, the temperature effect can be eliminated, leaving only the wavelength shift due to strain. This can be expressed by the following equation: (5)$$\Delta h=\left(\rho +h\right)\left(\frac{1}{\nu }+1\right)\left(\Delta \lambda_{h}-\Delta \lambda_{a}\right)/k\lambda ,$$
where $\Delta \lambda_{h}$ is the Δλ for hoop strain, and $\Delta \lambda_{a}$ is the Δλ for axial strain. Therefore, when modeling the strain of a reduced-wall elbow under different loads, only the internal pressure and bending moment loads need to be taken into account, not the temperature variation. Typically, the strain coefficient *k* of FBG is 1.19 pm/με [52]. By detecting the Bragg wavelength shift, the strain and wall thinning of the pipe elbow can be determined.

## 3. Simulation of a Pipe Elbow with Wall Thickness Reduction under Various Loads

In order to study the strain distribution of corroded bends under different loads, ANSYS software was used to build an FEM of a metal pipe elbow with decreasing wall thickness. The model simulates different degrees of corrosion with different thicknesses of the arch back of the pipe elbow and analyzes the strain distribution of the arch back of the pipe elbow under internal pressure and bending moment loads. Considering the geometrical symmetry of the elbow, only a quarter of the elbow is modeled, as shown in Figure 2.

The symmetry constraint is applied to the symmetry plane and axial profile of the elbow. It employs SOLID185 elements, and Table 1 summarizes the material properties and structural parameters. The parameters consist of material properties, including the elastic modulus of 206 GPa and a Poisson ratio of 0.3, and geometric properties, such as a pipe thickness of 20 mm, a bend radius of 3 in relation to the mean radius, a thinning length of 1.0 in relation to the initial diameter, a thinning depth of 0.534 in relation to the nominal thickness, and a thinning angle of 0.5 in relation to π. The meshing balances computational efficiency and computational accuracy, with the meshing being densest at the bends to obtain higher computational accuracy and sparser the farther away from the bends. In order to verify the correctness of the model, we built the model according to the parameters detailed in Ref. [53]. The model accounts for deformation in the “closing mode”, where the ends of the elbow move closer together during bending, as show in Figure 3. We calculated the relationship between the bending moment at the end of the pipe and the rotation angle, as shown in Figure 4.

The results show that the bending moment on the outside of the elbow increases rapidly as the rotation angle of the end of the elbow increases and then tends to stabilize, which is consistent with the results of the previous study [53], as shown in Figure 5.

### 3.1. Internal Pressure on the Inner Wall of the Pipe Elbow

To investigate the relationship between the variation of strain with the degree of corrosion of the pipe elbow under internal pressure loading, a hydrostatic internal pressure of 10 MPa was applied to the inner wall of the pipe elbow at room temperature, as shown in Figure 6. The red color mesh indicates the hydrostatic load applied to the grid points. For example, when the inner wall of the pipe elbow extrados is thinned by 50%, the axial and hoop strain contours of the bend are shown in Figure 7. The hoop tensile strain of the pipe elbow extrados shown in Figure 7a is about 0.02036, which is significantly larger than the axial strain of 0.00455 shown in Figure 7b. To find out the relationship between the change in the strain with the degree of corrosion of the pipe elbow, we simulated a pipe elbow subjected to internal hydrostatic pressure of 10 MPa stress on the inner wall of the elbow thinned from 0 to 95% in steps of 10%, where the percentage refers to the ratio of the corroded wall thickness to the original thickness. Figure 8 shows that both the hoop strain and axial strain on the outer side of the extrados of the elbow increase monotonically with decreasing wall thickness, especially with corrosion levels greater than 50%. This suggests that the strain on the arch back of the bend due to hydrostatic pressure in the early stages of corrosion is small and not easy to observe and that the strain is easily observable when it is already in the later stages of corrosion. In addition, in order to exclude temperature interference, the difference between hoop strain and axial strain is needed to characterize the change in the degree of corrosion. It can be seen that the difference between the two remains essentially constant over the range of 0 to 90% change in corrosion extent. Therefore, strain monitoring due to hydrostatic pressure cannot be used for early warning of bend corrosion monitoring.

### 3.2. Bending Moment on the Ending of the Pipe Elbow

To investigate the relationship between the change in strain on the exterior of the bent pipe and the degree of corrosion under moment loading, a force of 8 kN was applied to the end of the straight pipe to generate a bending moment. Concurrently, a fixed constraint was applied at the midpoint on the inner side of the bend, as illustrated in Figure 9.

The red arrow indicates the direction of the bending moment. The circumferential and axial strain distributions of the pipe elbow are shown in Figure 10a,b, respectively, when the thickness of the arch back of the pipe elbow is reduced by 50%. There is a significant tensile strain along the hoop direction on the outside of the bend, while there is a compressive strain along the axial direction. To characterize the relationship between strain and the degree of corrosion, simulations were conducted to explore how strain varies with corrosion under bending moment loads. The degree of corrosion is defined as the ratio of wall thickness before and after corrosion.

The simulation results are presented in Figure 11. As the degree of corrosion increases, the hoop strain decreases monotonically, while the axial strain increases monotonically, as depicted by the corresponding Bragg wavelength shift in Figure 12. This behavior occurs because, under bending moment loading, the compressive stress on the inner side of the pipe elbow decreases, while the tensile stress on the outer side increases, as shown in Figure 1, leading to a gradual decrease in hoop strain and a corresponding increase in axial strain. When the corrosion level reaches 90%, the progressive thinning of the pipe wall results in a collapse at the top of the pipe elbow, as shown in Figure 13. This collapse leads to a reduction in axial strain, while the Poisson effect leads to a more pronounced reduction in hoop strain. Despite the sudden drop in strain caused by the wall collapse, the strain difference between the two continues to increase monotonically. This observed trend aligns with the behavior of strain differences in bent metal plates of various thicknesses [51]. Importantly, the slope of the strain difference is positively correlated with the degree of corrosion under different bending moment loads, making it a reliable indicator for quantifying changes in the degree of corrosion.

When the corrosion level is between 1% and 10%, the change in the slope of the strain difference is relatively small. However, when monitoring with Bragg wavelength shifts, the circumferential strain leads to a wavelength shift ranging from −65.5 pm to −145.8 pm, while the axial strain results in a shift from −2025.3 pm to −1851.3 pm. The corresponding change in the slope of the Bragg wavelength difference is from 31.2 to 32.0. Despite the slight change in the slope of the strain difference, FBG sensors can still detect early-stage corrosion.

As the corrosion level increases, the slope change becomes more pronounced, making it easier to monitor corrosion through the strain difference. For instance, when the corrosion level increases from 50% to 80%, the slope of the Bragg wavelength shift accelerates, rising from 43.63 at 50% corrosion to 94.7 at 80%. When the corrosion level reaches 90%, the slope further increases significantly to 146.7.

This condition is similar to the previous experimental results where, at a wall-thinning percentage of 28%, the pipe elbow shows strain changes in different vibration modes due to collapse, thus affecting the frequency ratio. As shown in Figure 14, the m2 and n2 symmetric (SYM) mode of vibration indicates a significant collapse of the bend due to corrosion thinning. In addition, the m2, n3 asymmetric (ASYM) and the m1, n4 SYM modes show similar patterns.

### 3.3. Combined Internal Pressure and Bending Moment

In order to obtain the relationship between strain and corrosion degree of the bent pipe under complex loading, we superimpose the strain distribution under the effect of hydrostatic internal pressure and bending load according to the principle of mechanical superposition. The hoop and axial strains and their differences in the arch back of the bend are obtained, as shown in Figure 15.

It can be seen that, under the action of 10 MPa hydrostatic internal pressure and 8 kN bending moment load, the integrated hoop direction strain decreases slightly, while the integrated axial strain increases monotonically. The two oscillate at corrosion levels greater than 80%, which is caused by the collapse of the bend structure. The difference between the two increases monotonically with the degree of corrosion, and the amplitude of the change by the hydrostatic pressure will negligibly interfere. In conclusion, under different bending moments, the overall upward and downward shifts of the curves of the annular and axial strains change with the degree of corrosion of the bend. The difference between these strains increases with the degree of corrosion, showing a positive correlation. Although different loading conditions can influence the strain difference in actual working conditions, its slope remains positively correlated with corrosion levels. Therefore, the slope of the strain difference can serve as a reliable indicator for corrosion monitoring. Under combined loading conditions, the positive correlation between the slope of the strain difference and the degree of corrosion tends to be more pronounced compared to pure moment loading, as show in Figure 15. Figure 16 shows the change in the corresponding Bragg wavelength with the corrosion degree. For instance, when the corrosion level ranges from 1% to 10%, the circumferential strain causes the Bragg wavelength to shift from 430.6 pm to 394.9 pm, while the axial strain shifts it from −914.2 pm to −713.1 pm. Correspondingly, the slope of the Bragg wavelength difference increases from 26.3 to 28.3. Further observations reveal that the slope increases from 71.1 to 110.2 as the corrosion level rises from 70% to 80%, then from 110.2 to 168.2 as the corrosion level progresses from 80% to 90%, and finally jumps from 168.2 to 295.4 as corrosion reaches 90% to 96%. Observations indicate that the slope of the strain difference increases significantly at corrosion levels of 80% and 90%. This is because, at the 80% corrosion level, the thinning of the pipe wall makes it more susceptible to strain. Due to the Poisson effect, the hoop strain increases less compared to the axial strain, resulting in a sudden increase in the strain differential, which intensifies the increase in slope. When the corrosion level reaches 90%, further collapse occurs, causing a reduction in strain. The Poisson effect causes the hoop strain to decrease more rapidly than the axial strain, which further exacerbates the increase in the slope. Similar collapses may occur at different stages of corrosion due to the structure of the pipe elbow and material parameters, such as Young’s modulus and Poisson’s ratio. Despite these collapses, the strain differential continues to increase with corrosion. These collapses cannot be accurately validated through analytical solutions or experimental calibration, underscoring the significance of FEA methods.

## 4. Discussion

In order to fully assess the usefulness of this method, its applicability to different materials must be further discussed. Recent studies have shown that materials such as SS310, Ti-23Nb alloy, and Alloy 740 have excellent corrosion resistance [15,16]. However, even these corrosion-resistant materials are prone to corrosion after prolonged use. The changes in hoop strain, axial strain, and strain differential at pipe elbows are influenced by the physical properties of the material. For example, corrosion-resistant materials may exhibit strain changes over longer periods of time. In addition, differences in yield strength and toughness of different materials can lead to changes in the slope of the strain differential. Structural differences in pipe elbows can also affect strain changes. Notably, despite the differences in material properties, the effect of corrosion on strain change is consistent. Therefore, the slope of the strain difference remains a reliable indicator for monitoring the extent of corrosion in different materials. Even in the early stages (1% to 10%) of corrosion, where strain changes are minimal, the strain-induced Bragg wavelength shift under bending loads can be as much as 80 pm, and the strain-induced Bragg wavelength shift under combined bending moments and internal pressures can be as much as 36 pm. Due to the high wavelength accuracy of the FBG sensors, shifts as small as 10 pm are detected, making it possible to detect small strain changes caused by early-stage corrosion. The FBG sensor is able to detect small strain changes caused by early corrosion.

## 5. Conclusions

Our study highlights the significance of monitoring strain behavior in metal elbows under varying degrees of corrosion and complex loading conditions. Through finite element simulations and the principle of mechanical superposition, we have elucidated the relationship between strain changes and corrosion levels in metal elbows subjected to hydrostatic internal pressure and bending loads.

Our findings emphasize the importance of analyzing the difference between hoop and axial strains to accurately assess the corrosion degree of metal pipe elbows while minimizing the influence of temperature variations. Under hydrostatic internal pressure loading conditions, our results indicate that both hoop and axial strains on the outer side of the pipe elbow increase as wall thickness decreases due to corrosion, particularly at corrosion levels exceeding 50%. However, the difference between hoop and axial strains due to hydrostatic pressure changes modestly during the early stages of bend corrosion, limiting its utility for early corrosion detection.

Furthermore, under bending moment loading, the hoop strain gradually decreases as corrosion progresses, while the axial strain gradually increases. The difference between the two strains consistently shows an upward trend as the degree of corrosion increases.

By superimposing strain distributions under hydrostatic internal pressure and bending load, we observe that axial strain increases monotonically with corrosion degree, while hoop strain shows a more gradual change. The difference between the strains increases with the increase in corrosion levels, especially under bending moment load. However, under the combined effects of hydrostatic pressure and bending moment, idealized assumptions, material properties, and boundary conditions may lead to discrepancies between simulation results and actual conditions. As a result, relying solely on the strain difference curve to assess corrosion progression may not be accurate. Instead, the positive correlation between the slope of the strain difference and the degree of corrosion serves as a crucial parameter for evaluating the corrosion level in pipe elbows.

In conclusion, this study provides valuable insights into the complex interactions between strain behavior and corrosion effects in metal elbows under different loading conditions. The established relationship between strain properties and corrosion extent provides a solid theoretical foundation for future experimental studies in this field and provides guidance for the development of effective monitoring and maintenance strategies for corroded structural components. Notably, structural parameters, material properties, and boundary conditions may lead to discrepancies between simulation results and actual conditions. Consequently, future research will focus on experimental validation to accurately assess the impact of these discrepancies on model predictions. This approach will offer valuable insights into corrosion monitoring for practical applications.

## Figures and Tables

**Figure 1 micromachines-15-01098-f001:**
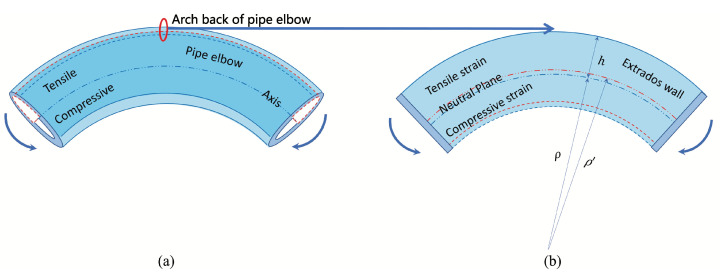
Illustration of (**a**) a corroded pipe elbow under the bending moment and (**b**) the bending wall at the convex part of the pipe elbow with the curvature radius ρ and the wall thickness 2h.

**Figure 2 micromachines-15-01098-f002:**
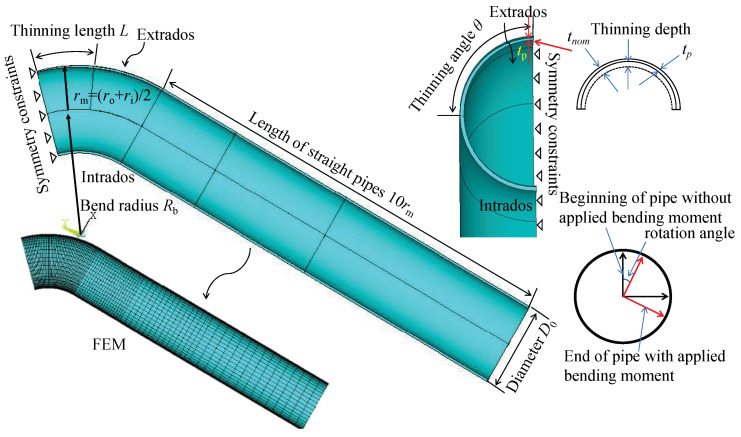
Dimensions of a wall-thinned pipe elbow and finite element mesh.

**Figure 3 micromachines-15-01098-f003:**
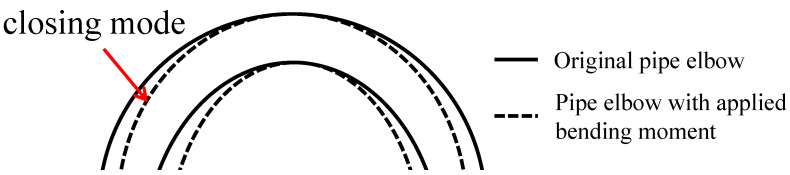
Closing mode of the pipe elbow.

**Figure 4 micromachines-15-01098-f004:**
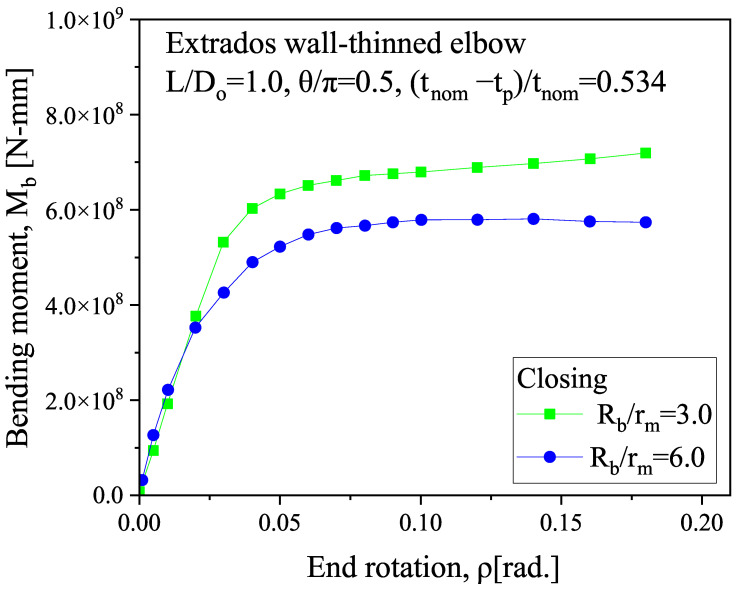
Relationship between the bending moment and end rotation at the extrados wall-thinning of the elbow.

**Figure 5 micromachines-15-01098-f005:**
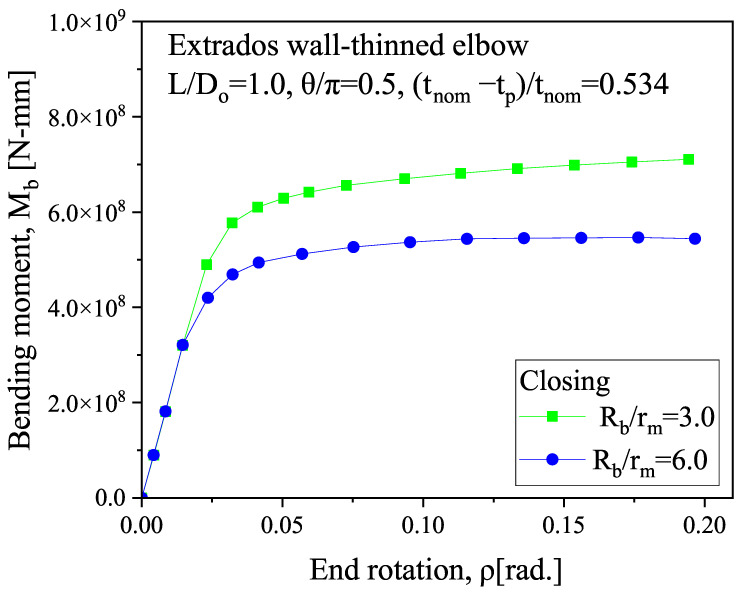
Relationship between the bending moment and end rotation at the extrados wall-thinning from previous results [53].

**Figure 6 micromachines-15-01098-f006:**
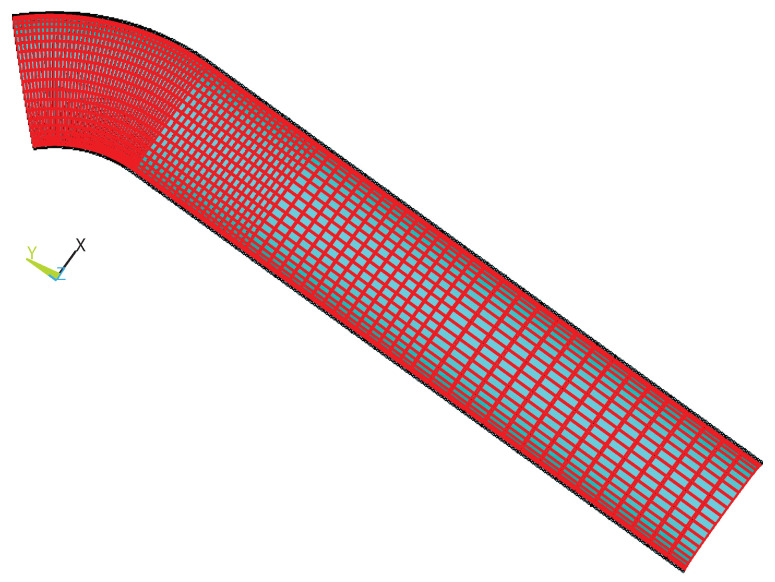
Inner wall of the pipe elbow under 10 MPa pressure.

**Figure 7 micromachines-15-01098-f007:**
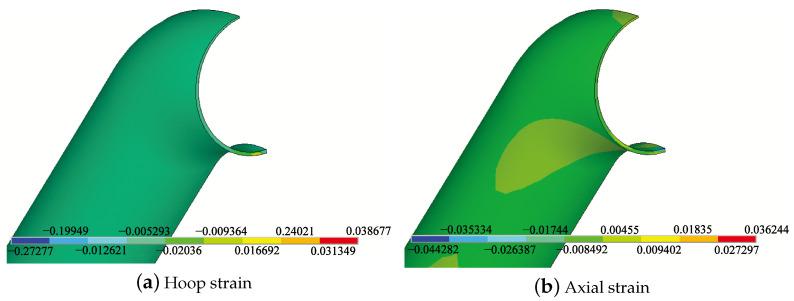
Hoop (**a**) and axial (**b**) strain distribution of the pipe elbow with 50% thickness thinning under 10 MPa internal pressure.

**Figure 8 micromachines-15-01098-f008:**
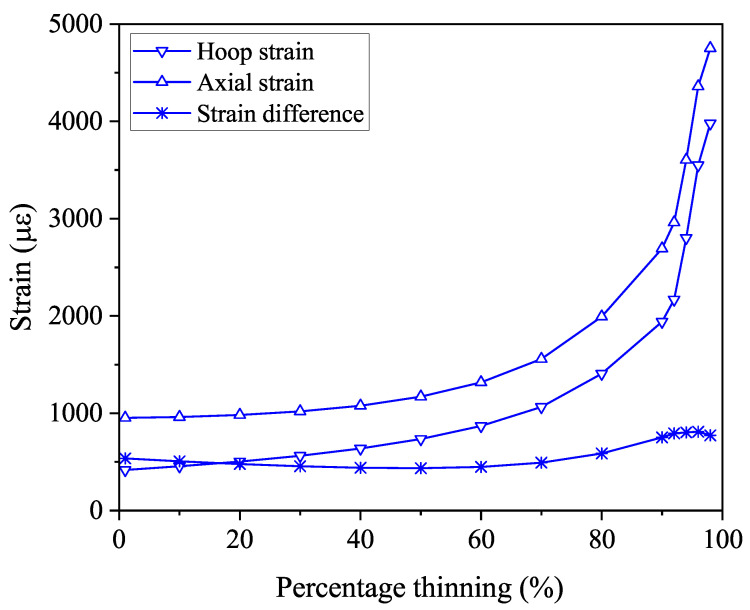
Hoop strain, axial strain, and the difference between them for the pipe elbow extrados variation against the percentage thinning of the pipe elbow under 10 MPa internal pressure.

**Figure 9 micromachines-15-01098-f009:**
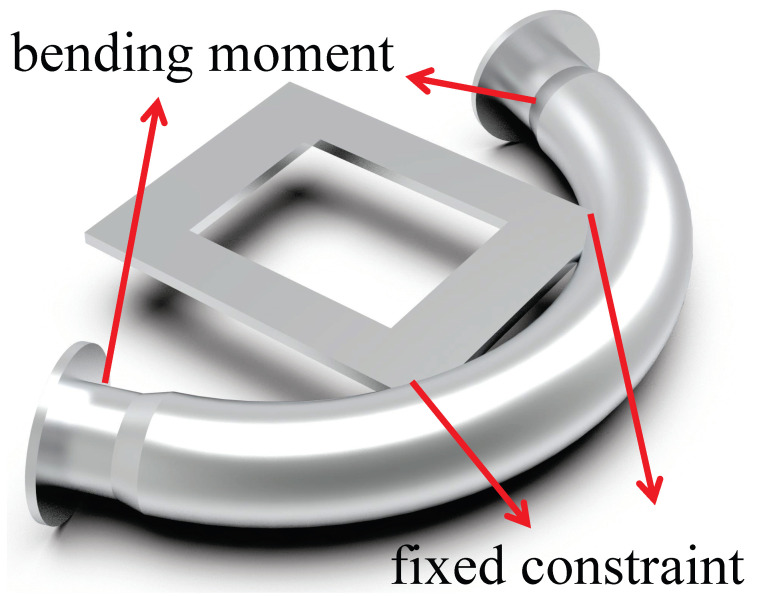
The pipe elbow with the bending moment and displacement constraints at the intrados of the pipe elbow.

**Figure 10 micromachines-15-01098-f010:**
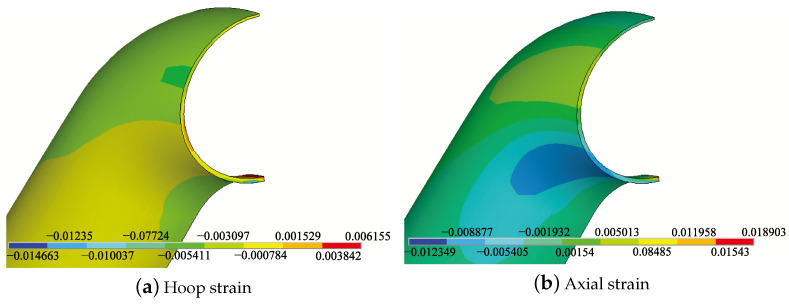
Hoop (**a**) and axial (**b**) strain distributions on the pipe elbow with 50% thickness thinning under bending moment load.

**Figure 11 micromachines-15-01098-f011:**
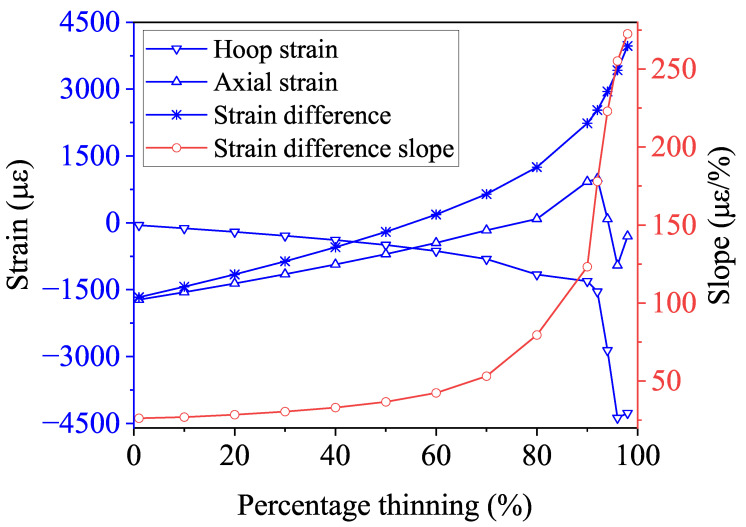
Hoop strain, axial strain, and strain difference between them for the pipe elbow extrados variation against the percentage thinning of the pipe elbow under the bending moment.

**Figure 12 micromachines-15-01098-f012:**
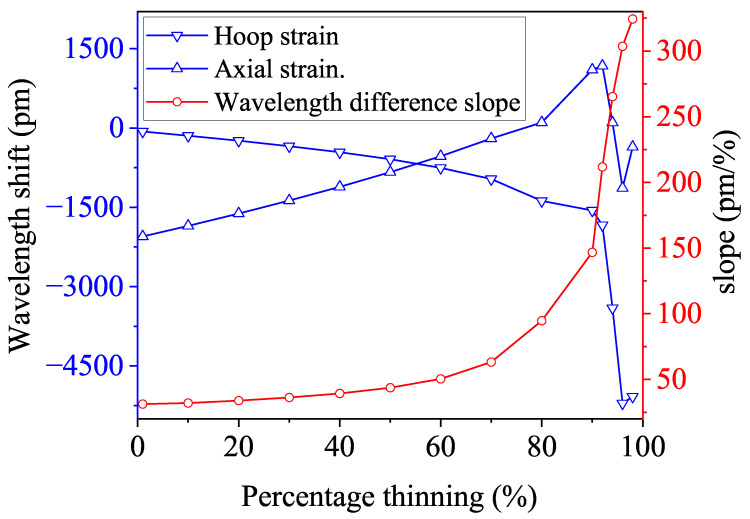
The Bragg wavelength slopes with the percentage of the pipe elbow thinning under the bending moment.

**Figure 13 micromachines-15-01098-f013:**
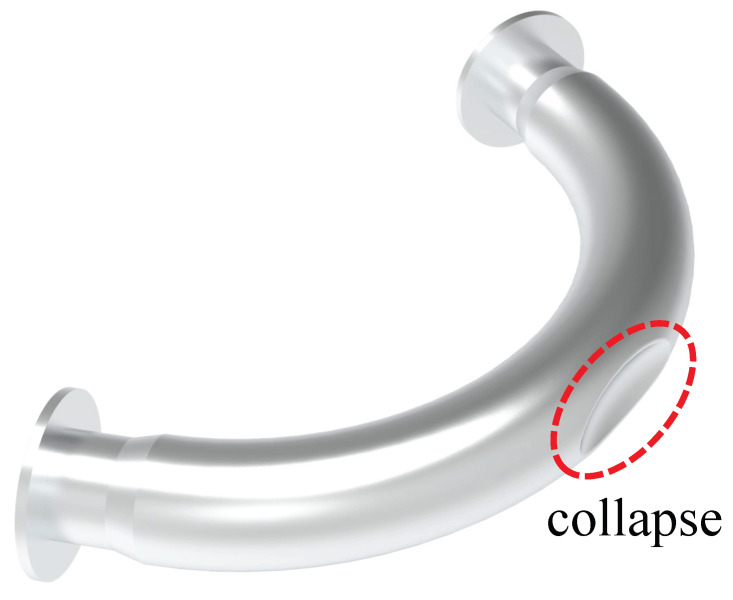
Collapse of pipe elbows when wall thinning is too high.

**Figure 14 micromachines-15-01098-f014:**
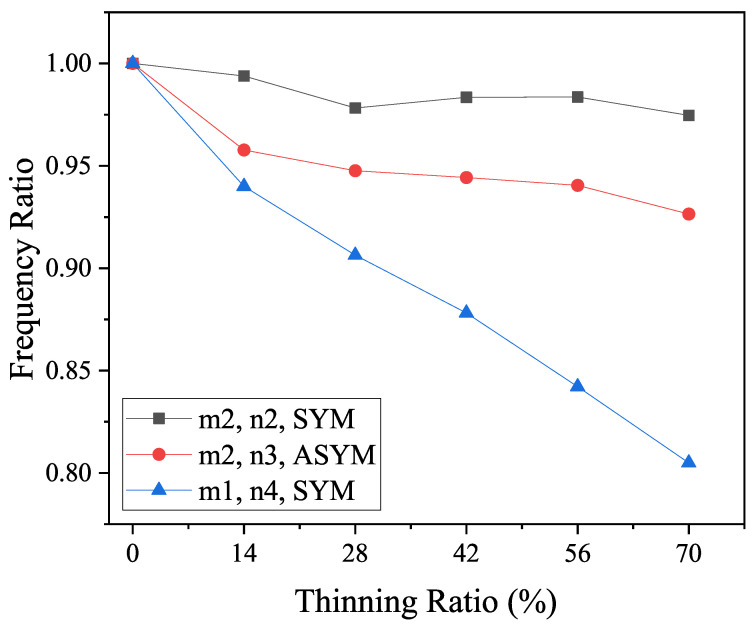
From the experimental modal analysis, the ratio of the intrinsic frequency of the thinned elbow to the intrinsic frequency of the intact elbow for different mode shapes varies with the percentage of thinning of the pipe elbow [27].

**Figure 15 micromachines-15-01098-f015:**
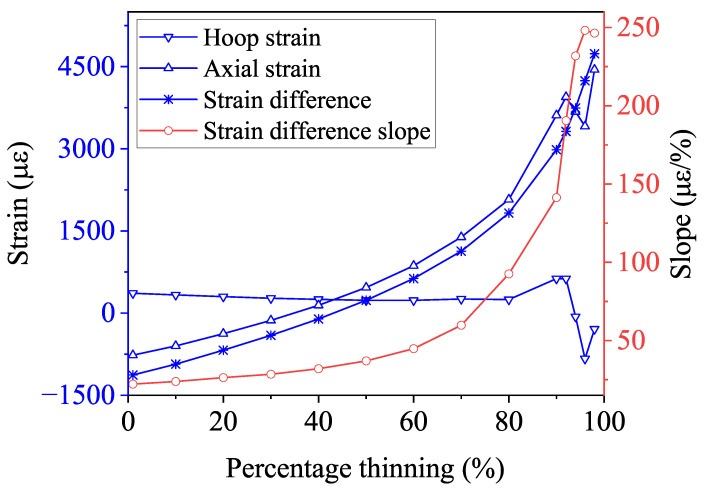
Hoop strain, axial strain, and strain difference between them of the pipe elbow extrados variation against the percentage thinning of the pipe elbow under combined internal pressure and the bending moment.

**Figure 16 micromachines-15-01098-f016:**
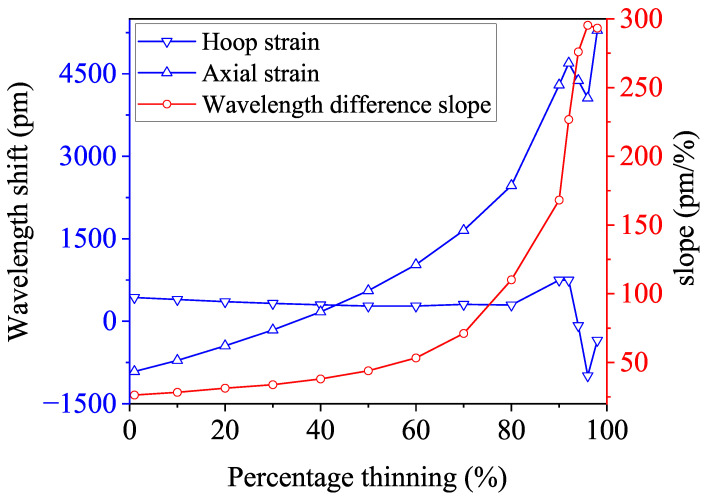
The Bragg wavelength slopes with the percentage of the pipe elbow thinning under combined internal pressure and the bending moment.

**Table 1 micromachines-15-01098-t001:** Parameters of the pipe elbow.

Illustration	Quantity
Elastic module	206 GPa
Poisson ratio	0.3
Thickness of pipe	20 mm
Bend radius, $R_{b}/r_{m}$	3
Thinning length, $L/D_{0}$	1.0
Thinning depth, $\left(t_{nom}-t_{p}\right)/t_{nom}$	0.534
Thinning angle, θ/π	0.5

## Data Availability

Data available on request from the authors.

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
