# Peer review of "Pipeline Elbow Corrosion Simulation for Strain Monitoring with Fiber Bragg Gratings"

_micromachines, 2024, doi:10.3390/mi15091098_

Round 1
Reviewer 1 Report
Comments and Suggestions for Authors
The topic of predicting pipeline corrosion through strain monitoring using the FBG method is interesting. However, there are still some questions that need to be addressed before publication can be considered. For example, the reviewer didn't find a description of Figure 4 in the manuscript. In addition, did the authors take into account the effect of different materials (such as steel and titanium) on the simulation results? Additional discussion is needed here, some of the references (Corrosion Science, 2021, 184: 109350; Materials Characterization, 2023, 197: 112647) could be helpful. Finally, the reviewer is curious about the differences between the simulation and the actual situation. Please discuss in detail.
Reviewer 2 Report
Comments and Suggestions for Authors
See attached Word document

Needs to be proof read
Author Response
请参阅附件。

Round 2
Reviewer 2 Report
Comments and Suggestions for Authors
Suggestions have been taken and amendments have been made
Comments on the Quality of English LanguageGrammatical errors are still seen.